

# Adaptive and degenerative evolution of the S-Phase Kinase-Associated Protein 1-Like family in *Arabidopsis thaliana*

Zhihua Hua[1] and Zhenyu Gao[1,2]

[1] Department of Environmental and Plant Biology and Interdisciplinary Program in Molecular and Cellular Biology, Ohio University, Athens, OH, USA

[2] State Key Laboratory of Rice Biology, China National Rice Research Institute, Hangzhou, China

## ABSTRACT

Genome sequencing has uncovered tremendous sequence variation within and between species. In plants, in addition to large variations in genome size, a great deal of sequence polymorphism is also evident in several large multi-gene families, including those involved in the ubiquitin-26S proteasome protein degradation system. However, the biological function of this sequence variation is yet not clear. In this work, we explicitly demonstrated a single origin of retroposed *Arabidopsis Skp1-Like* (*ASK*) genes using an improved phylogenetic analysis. Taking advantage of the 1,001 genomes project, we here provide several lines of polymorphism evidence showing both adaptive and degenerative evolutionary processes in *ASK* genes. Yeast two-hybrid quantitative interaction assays further suggested that recent neutral changes in the *ASK2* coding sequence weakened its interactions with some F-box proteins. The trend that highly polymorphic upstream regions of *ASK1* yield high levels of expression implied negative expression regulation of *ASK1* by an as-yet-unknown transcriptional suppression mechanism, which may contribute to the polymorphic roles of Skp1-CUL1-F-box complexes. Taken together, this study provides new evolutionary evidence to guide future functional genomic studies of SCF-mediated protein ubiquitylation.

## INTRODUCTION

Proteins play fundamental roles in driving life processes by sensing diverse environmental cues, catalyzing biochemical reactions, monitoring the stability of genetic materials, and combating abiotic and biotic stresses. In addition, they are believed to be the only molecules capable of mechanical movement in any organism. To accomplish these diverse roles, not only is protein synthesis precisely controlled, but the structure, activity, and turnover of each protein is also sophisticatedly regulated in a temporal and spatial manner. The ubiquitin-26S proteasome system (UPS) is the primary degradative machinery for rapidly modulating protein content in eukaryotic cells. Given the power of its selective turnover of numerous intracellular proteins, the UPS plays an essential regulatory role in controlling cell cycle progression, signal transduction, gene expression regulation,

Corresponding author
Zhihua Hua, hua@ohio.edu

genome stability, and many other cellular processes (*Finley et al., 2012*; *Vierstra, 2009*; *Yau & Rape, 2016*). This function has been demonstrated to be particularly important in plants, as evidenced by the extremely large expansion of several gene superfamilies that encode plant UPS members (*Vierstra, 2009*).

Among these members, S-Phase Kinase-Associated Protein 1 (Skp1), F-box proteins, Cullin 1, and RING box protein 1 compose the largest group of ubiquitin-ligase complexes in plants, termed SCF complexes, which target protein substrates for ubiquitylation and subsequent turnover by the 26S proteasome (GO:0019005) (*Hua & Vierstra, 2011*). Genetic, genomic, evolutionary, and biochemical analyses have shown that the *F-box* multi-gene superfamily encodes a substrate receptor that determines the specificity of the SCF complex, while the Skp1 protein family functions as an adaptor to bridge the variable F-box proteins to the N-terminus of Cullin 1 to assemble a holo-ubiquitin ligating enzyme (*Gagne et al., 2002*; *Hua & Vierstra, 2011*; *Yang et al., 1999*; *Zheng et al., 2002*). Although a handful of F-box proteins have been functionally shown to target the degradation of proteins involved in the cell cycle, circadian rhythms, photomorphogenesis, pathogen defense, hormone signaling, and plant reproduction, many recently duplicated and species- or lineage-specific members remain uncharacterized (*Hua et al., 2011*). Given that the size of the *F-box* superfamily is species-specific and is often not correlated with the complexity of plant species, a genomic drift evolutionary mechanism has been postulated to explain the random size drift of the *F-box* gene superfamily in plants (*Hua et al., 2011*; *Nozawa, Kawahara & Nei, 2007*; *Xu et al., 2009*). The high sequence polymorphism of lineage specific *F-box* genes and their enrichment of transcriptional suppression-related epigenetic modifications further support this hypothesis (*Hua et al., 2013*). However, this does not preclude the existence of some young *F-box* genes that play a lineage specific role in plant adaptation (*Gagne et al., 2002*; *Shabek & Zheng, 2014*; *Yang et al., 2008*). Unfortunately, it remains difficult to find these members both experimentally and theoretically, in part due to the large size of this group and the low/no expression of most lineage specific *F-box* genes (*Hua et al., 2013*).

Similar to the *F-box* gene superfamily, the *Skp1* family has also expanded significantly in land plants. While there is only one single Skp1 protein encoded in yeast and human genomes, the genomes of *Arabidopsis thaliana* and *Oryza sativa* contain 21 and 32 annotated *Skp1* loci, respectively (*Kong et al., 2007*). Cross-kingdom evolutionary studies have suggested that the plant *Skp1* genes are also rapidly evolving through a birth-and-death evolutionary mechanism (*Kong et al., 2004*, *2007*). However, unlike many inactive *F-box* genes (*Hua et al., 2011*; *Kuroda et al., 2012*), 20 out of 21 *A. thaliana Skp1-Like* (*ASK*) genes are transcribed in at least one out of six tissues/organs examined, including seedlings, roots, stems, leaves, inflorescences, and siliques (*Kong et al., 2004*; *Zhao et al., 2003*), suggesting that most *ASK* genes are active. Phylogenetic analysis further implied that all plant *Skp1* genes shared one common ancestor, although evolutionary rates of individuals are highly heterogeneous. Therefore, it has been inferred that some moderately and rapidly evolving members might have lost their original functions and/or gained new functions (*Kong et al., 2004*). Despite the sequence diversity of ASK proteins, a recent biochemical study showed that all ASK proteins retained the
biochemical function of their ancestor Skp1 protein for interacting with F-box proteins (*Kuroda et al., 2012*).

The fact that all ASK proteins interact with an F-box protein implied that their sequences are not sufficiently diverged from their ancestor sequence for their original biochemical function to have been lost. However, this rapid evolution has dramatically diversified the sequences of plant *Skp1* genes, making it challenging to uncover the true phylogenetic relationships among distantly related plant species. Indeed, to avoid the effects of long-branch attraction, type II *Skp1* genes, which carry multiple introns in various positions, unlike type I *Skp1* genes that contain only one or none introns, were excluded in a previous phylogenetic analysis of *Skp1* genes (*Kong et al., 2007*). Improved understanding of the evolutionary mechanisms of the *Skp1* gene family may aid further exploration of the functions of many unknown SCF complexes. To date, the genome sequences of three Arabidopsis species, *A. thaliana*, *A. lyrata*, and *A. halleri*, which split 5–10 million years ago (mya) (*Hu et al., 2011*; *Koch & Kiefer, 2005*), have been obtained (The Arabidopsis Information Resource (TAIR), V10; Phytozome V12), and >1,000 individual *A. thaliana* accessions have been sequenced (*1001 Genomes Consortium, 2016*). These datasets can allow us to further fine-tune the phylogenetic relationships and fixation processes of rapidly-evolving genes in plants, which may help better define their functional constraints. Because the diverse functions of the SCF complexes are primarily determined by the large *F-box* gene family along with the *Skp1* gene family, in this work we analyzed the short evolutionary history of the *Skp1* genes within and between Arabidopsis species in order to uncover important evolutionary patterns in SCF regulatory pathways. Our new evidence suggests that the *ASK* genes are under both adaptive and degenerative evolutionary processes.

## MATERIALS AND METHODS

### Identification of *Skp1* genes in *A. lyrata* and *A. halleri*

The full set of Skp1 seed sequences that encompass a 70–86 amino acid core Skp1 domain were retrieved from Pfam (PF01466, Version 27, http://pfam.xfam.org) and used as query in a BLASTp search (*Altschul et al., 1990*) against the annotated proteome of each species, which was retrieved from Phytozome (http://phytozome.jgi.doe.gov/; *A. lyrata* V2 and *A. halleri* V1.1). The presence of Skp1 and any additional protein–protein interacting domains in each full-length hit sequence were further confirmed by hmmscan (http://hmmer.org) against the Pfam-A database (Pfam 27, http://pfam.xfam.org). To identify a complete list of *Skp1* genes in each species, a previously developed sequence similarity-based annotation algorithm, called Closing Target Trimming (*Hua & Early, 2018*; *Hua et al., 2011*), was also used to search the genomes for any new *Skp1* loci that may not have been annotated.

### Sequence alignment and phylogenetic analysis

Instead of manual adjustment and artificial deletion of ambiguous alignment as reported in the previous studies (*Kong et al., 2004*, *2007*; *Zhao et al., 2003*), two Skp1 protein sequence alignments were obtained by MUSCLE (*Edgar, 2004*) and MAFFT

(*Katoh, Rozewicki & Yamada, 2017*), and then used to make a consensus alignment by Trimal (-conthreshold 0.5) (*Capella-Gutierrez, Silla-Martinez & Gabaldon, 2009*). The resulting alignment was used to generate a maximum likelihood (ML) phylogenetic tree by RAxML (Version 8.1; *Stamatakis, 2014*) with the PROTGAMMAJTT substitution model. The statistical significance was evaluated with 1,000 bootstrap replicates using a rapid bootstrap analysis.

## Birth and death of the *Skp1* genes in the Arabidopsis genus

Gene duplication and loss events were inferred by reconciling the ML gene tree with the species tree using Notung (version 2.9) (*Chen, Durand & Farach-Colton, 2000*).

## Gene structure analysis and reference sequence retrieval

The number of introns in each *Skp1* gene was counted based on the Generic Feature Format (GFF3) file from each genome project. According to the chromosomal coordinates, the upstream and downstream regions of a *Skp1* gene, which are 500 nucleotides upstream of the start codon and downstream of the stop codon, respectively, were retrieved from the genomes of *A. thaliana* (TAIR V10; www.arabidopsis.org) and *A. lyrata* (V2; *Rawat et al., 2015*). The coding sequence (CDS) of an *ASK* gene within the Col-0 reference genome and its *A. lyrata* ortholog was retrieved from the annotated transcriptomes of *A. thaliana* (TAIR V10) and *A. lyrata* (V2; *Rawat et al., 2015*), respectively.

## Sequence assembly for polymorphism analysis

To assemble an *ASK* allelic sequence, single-nucleotide polymorphic (SNP) alleles (Phred quality score ≥25) within the coding and non-coding flanking sequences were first retrieved from each Arabidopsis accession (http://1001genomes.org). In total, 774 accessions were selected (*Cao et al., 2011*; *1001 Genomes Consortium, 2016*; *Long et al., 2013*; *Schmitz et al., 2013*). Based on their co-ordinates, both variant and the Col-0 reference SNP alleles were used to substitute the nucleotides in a reference sequence to assemble two allelic sequences. Only when the new Col-0 allelic sequence was 100% identical to the reference sequence was the variant allele considered to be assembled correctly. To assemble an outgroup sequence, an amino acid sequence alignment of an ASK protein and its *A. lyrata* ortholog was obtained by MAFFT (*Katoh, Rozewicki & Yamada, 2017*) and used to guide the assembly of a nucleotide sequence of the outgroup. The sites introducing gaps in the reference *ASK* sequence were removed.

## Determination of tandemly duplicated *Skp1* genes

Two *Skp1* genes were determined to be tandemly duplicated if they were both separated by ≤5 genes and located within 10 kb.

## Clustering analysis

Sequences were clustered using Heatmap.2 (dist method = "manhattan," hclust method = "word.D") in R (*R Core Team, 2017*) to demonstrate similar evolutionary constrains of mutations as described previously (*Hua, Doroodian & Vu, 2018*).

## Expression data resources

The RNA-Seq data for leaf or seedling transcriptomes from 144 to 19 different *A. thaliana* accessions were retrieved from the projects published by *Schmitz et al. (2013)* and *Gan et al. (2011)*, respectively. Microarray expression data of 79 samples collected from eight different tissues/organs throughout the *A. thaliana* Col-0 life cycle were downloaded from http://jsp.weigelworld.org/expviz/expviz.jsp.

## Cross species test of neutral evolution

The orthologous pairing of a *Skp1* gene between *A. thaliana* and *A. lyrata* was determined by OrthoMCL (*Li, Stoeckert & Roos, 2003*) and used to examine its neutral evolutionary process as previously described (*Nekrutenko, Makova & Li, 2002*), with minor modifications. The nucleotide sequences of each pair were aligned based on the protein sequence alignment obtained by T-Coffee (*Taly et al., 2011*) and used as an input file to run the codeml program from the PAML4 package (*Yang, 2007*) twice, with the $Ka/Ks$ ratio either fixed at 1 or free. The ML values ML1 and ML2 from the two runs were collected to calculate the likelihood ratio as LR = 2(lnML1 – lnML2). If LR is less than 2.71 (5% significance for χ2 distribution with one degree of freedom) (*Yang, 2007*), the $Ka/Ks$ ratio is considered not significantly different from 1, that is, the *Skp1* gene is under a neutral evolutionary process.

## Molecular cloning and yeast two-hybrid analysis

The CDSs of 15 selected known *F-box* genes and *ASK1/2* were PCR amplified from cDNA clones that were obtained from the Arabidopsis Biological Resource Center (https://abrc.osu.edu) and ligated in-frame to the 3′-end of GAL4-BD and GAL4-AD CDSs present in the yeast two-hybrid vectors, pGBK-T7 (bait) and pGAD-T7 (prey), respectively. The resulting bait and prey vectors were separately transformed into the haploid yeast strains, AH109 and Y187, respectively, which were subsequently mated according to the pairwise interaction combinations.

For yeast growth interaction assays, the number of mated diploid yeast cells were normalized and diluted with sterile water in series to an $OD_{600}$ of 0.8, 0.4, 0.2, and 0.1. 5 µL of yeast cells from each concentration were then spotted on either a quadruple synthetic dropout medium (SD-Leu-Trp-Ade-His) containing X-α-gal (40 µg/mL) for interaction assays, or on a double synthetic dropout medium (SD-Leu-Trp) as a growth control. To quantify the interaction strength, the intensity of yeast growth from the scanned images was calculated using ImageQuant version 5.2 (GE Healthcare, Chicago, IL, USA). Each interaction signal on SD-Leu-Trp-Ade-His+X-α-gal medium was normalized to that detected on SD-Leu-Trp medium.

For the β-galactosidase activity assay, six to 10 mated yeast colonies grown on SD-Leu-Trp medium were freshly harvested and resuspended in 0.5 mL of Z buffer (50 mM sodium phosphate, 10 mM potassium chloride, two mM magnesium sulfate, pH 7.0) in a two mL deep-well plate. 0.1 mL of resuspended yeast cells were further aliquoted into a new two mL deep-well plate and used for β-galactosidase activity according to

*Miller (1972)*. Relative β-galactosidase activities were calculated based on the method described previously (*Hua & Kao, 2006*). In total, two independent replicates were assayed.

## RESULTS

### Identification of *Skp1* genes in three closely related Arabidopsis species

The list of 21 *A. thaliana ASK* genes, which contain 19 Type I (*ASK1-19*) and 2 Type II (*ASK20* and *21*) *Skp1* genes, has been well annotated in previous studies (*Farras et al., 2001*; *Kong et al., 2004*, *2007*; *Zhao et al., 2003*). However, a full list of *Skp1* genes has not been reported in *A. lyrata* or *A. halleri*. To identify a comprehensive list of *Skp1* genes in these two Arabidopsis genomes, BLASTp (*Altschul et al., 1990*) and hmmscan (http://hmmer.org) were first applied to search the available genome annotations (*A. lyrata* V2 (*Rawat et al., 2015*), *A. halleri* V1.1 (Phytozome V12)). In total, 17 and 11 loci were identified that encode a Type I Skp1 protein, and four and three loci were discovered to encode a Type II Skp1 protein, in *A. lyrata* and *A. halleri*, respectively (File S1). After a subsequent sequence-similarity based Closing Target Trimming search (*Hua & Early, 2018*; *Hua et al., 2011*), no additional hits were identified in each genome. The relatively long length of the Skp1 domain, the low number or absence of introns in the *Skp1* loci, and the small size of the *Skp1* family, may facilitate the annotation of *Skp1* genes in genomes. Therefore, in total 21, 21, and 14 *Skp1* loci are present in *A. thaliana*, *A. lyrata*, and *A. halleri*, respectively. The size variation among these closely related Arabidopsis species indicates that the *Skp1* family is, like the *F-box* gene superfamily, under a rapid birth-and-death evolutionary process.

### Phylogenetic analysis of the *Skp1* genes in a short evolutionary history

Since the sequences of Type II *Skp1* genes are significantly diverged from Type I *Skp1* genes, and most *Skp1* genes are Type I (*Kong et al., 2007*), hereafter we focused on the evolutionary study of the Type I group. To understand the birth-and-death process of the *Skp1* family in a short evolutionary history, we performed a phylogenetic analysis using an improved sequence alignment approach. Since manual adjustment and artificial deletion of ambiguous alignment sites are not always reproducible, Trimal (*Capella-Gutierrez, Silla-Martinez & Gabaldon, 2009*) was used to remove poorly aligned regions automatically. In addition, a consensus result was obtained from MUSCLE (*Edgar, 2004*) and MAFFT (*Katoh, Rozewicki & Yamada, 2017*) sequence alignments to improve accuracy (see "Materials and Methods"). The resulting alignment not only significantly reduced gaps and mis-matched sites, but also retained a reproducible result with 95 ± 5% of the full length Skp1 protein sequences being aligned (Fig. S1), maximizing the sequence length and variable sites essential for a good phylogenetic analysis (*Nei & Kumar, 2000*). As a proof of concept, an ASK2-rooted ML tree generated based on the 19 aligned ASK protein sequences showed a compatible topology to the one reported previously (*Kong et al., 2007*) (Fig. 1A). However, unlike the previous tree where intronless and intronic *ASK* genes are intermingled (*Kong et al., 2007*),

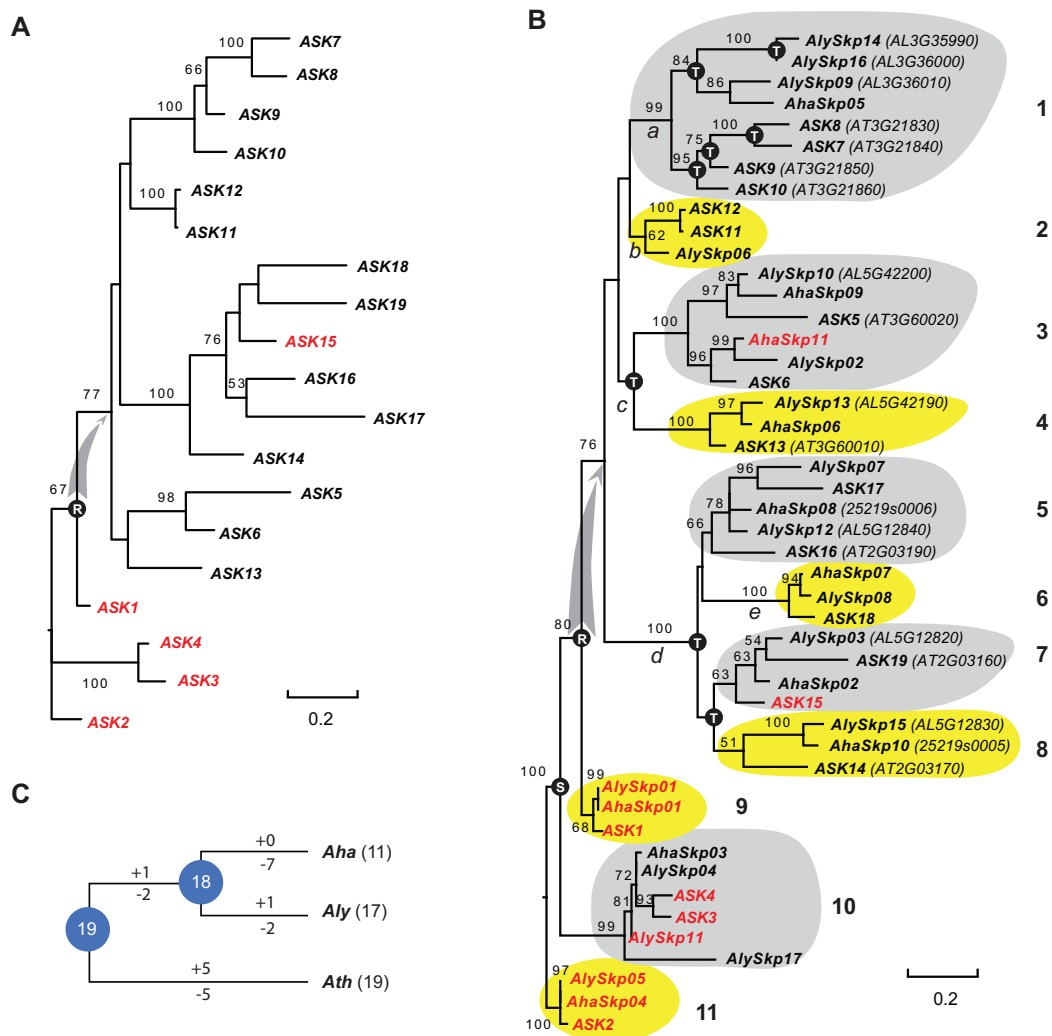

**Figure 1 A short evolutionary history of the Arabidopsis *Skp1* genes within the Arabidopsis genus.**
(A) Phylogenetic relationships of the *ASK* members. Intronic genes are highlighted in red. Scale bar, average substitutions per site. (B) An improved phylogenetic analysis reveals one single origin of retroposed *Skp1* genes in the Arabidopsis genus. *a–e*, 5 ancient retroposed loci produced by the transcripts of the highly expressed ancestor locus of *ASK1*. 1–11, 11 clades shaded with dark and gray color showing clear orthologous relationships among the three Arabidopsis species. The tree was generated by a maximum likelihood method. Statistical significance equal to or greater than 50% of 1,000 bootstrap resamplings is indicated in each node. S, segmental duplication; T, tandem duplication; R, retroposed duplication. The ID number from the original annotation is indicated if a *Skp1* gene is duplicated through tandem duplications or described in File S1. Intronic *Skp1* genes and scale bar are described as in (A). Species abbreviations: *Aha*, *A. halleri*; *Aly*, *A. lyrata*; *Ath*, *A. thaliana*. (C) Birth-and-death history of *Skp1* genes in the Arabidopsis genus. + and − indicate gain and loss of *Skp1* members, respectively. Species abbreviations are as in (B).

all intron-containing *ASK* genes (except for *ASK15*, whose intron was gained after duplication (*Kong et al., 2007*)) were clustered at the base of the tree (Fig. 1A). This result better explains a single origin of the intronless *ASK* genes, which were duplicated through retroposition from a highly expressed *ASK* gene, most likely the ancestor of *ASK1*.
Subsequently, a ML tree rooted to ASK2 was generated based on the consensus protein sequence alignment of the 47 Arabidopsis Skp1 protein sequences by RAxML (*Stamatakis, 2014*). Encouragingly, the resulting phylogenetic tree showed a similar topology to that obtained for the *ASK* genes, and the 47 *Skp1* genes from the three Arabidopsis species were intermingled in 11 clusters (Fig. 1B). Therefore, most Arabidopsis *Skp1* genes were duplicated at least 5–10 mya, at the time when the three Arabidopsis species split (*Hu et al., 2011*; *Koch & Kiefer, 2005*). Similarly to the *ASK* genes, all intronic Arabidopsis *Skp1* genes were clustered at the base of the tree, while the remaining intronless *Skp1* genes were clustered into one big clade, suggesting a common role of retroposition in the expansion of the *Skp1* family in Arabidopsis. Among these, three ancestor loci (Fig. 1B, nodes *a*, *c*, and *d*) were likely duplicated through retroposition, with each likely further undergoing tandem duplication events to yield the current *Skp1* members. Two *Skp1* clades (Fig. 1B, clades 2 and 6) were likely the direct product of a retroposition event. Therefore, the mRNAs produced by the highly expressed *ASK1* ancestor were likely retrotransposed to five ancestor loci in total (Fig. 1B, nodes *a–e*). This phylogenetic tree also showed a clear duplication event between the *ASK1* clade and the *ASK4* clade before the split of three Arabidopsis species. Although *Kong et al. (2007)* first reported the contribution of segmental duplication in duplicating the *ASK1* and *ASK4* loci, their phylogenetic tree did not reflect the direct connection between these two genes (*Kong et al., 2007*).

We further reconciled a gene tree based on this ML tree. Along with a species tree, we detected significant variance of birth/death rates between each species (Fig. 1C). While *A. halleri* lost 7 *Skp1* loci after a recent split from *A. lyrata*, *A. thaliana* has gained and lost five loci each from the 19 common Arabidopsis *Skp1* gene ancestors. The birth/death rate of *Skp1* genes in *A. lyrata* was intermediate among the three species; it gained and lost 2 and 4 *Skp1* genes, respectively, following the split from *A. thaliana*. Such a significant size variation even within a short evolutionary history implied that some ancestral retroposed *Skp1* loci resided in a hot spot of tandem duplications, which contributed to the differential sizes of the *Skp1* family among the three Arabidopsis species. For example, at the ancestor "*a*" locus, 4 and 3 *Skp1* genes were gained through tandem duplications within the past 5–10 million years in *A. thaliana* and *A. lyrata*, respectively (Fig. 1B).

## Low evolutionary constraints of intronless *ASK* genes

The phylogenetic tree revealed a clear orthology relationship between the *Skp1* genes of the three Arabidopsis species. To further demonstrate this relationship, we applied OrthoMCL (*Li, Stoeckert & Roos, 2003*) to identify 14 *Skp1* orthologous groups, among which 18 *ASK* genes have been partnered with one *A. lyrata* *Skp1* orthologous gene (Table S1). This clear orhology relationship allowed us to examine evolutionary constraints on the sequence divergence of *ASK* genes. We primarily applied the method of *Nekrutenko, Makova & Li (2002)* to test whether the *Ka*/*Ks* ratio (i.e., ω) of an *ASK* gene is significantly diverged from 1, which indicates between-species neutral nucleotide divergence. Surprisingly, nine out of 14 (64%) intronless *ASK* genes were detected to be under a neutral evolutionary process, while none of the four intronic *ASK* genes belongs to this category (Table 1), suggesting that the former group has lower functional

**Table 1 Maximum likelihood test of neutral evolution by comparing neutral evolutionary model (ML1: dN/dS = 1) and non-neutral evolutionary model (ML2: free dN/dS value) of *ASK* genes.**

| Ath Skps | Aly Skps | dS | dN | dN/dS | lnML1 | lnML2 | 2l nML | Selection* |
|----------|----------|-----|-----|-------|--------|--------|--------|------------|
| ASK2 | AlySkp05 | 0.2 | 0.0 | 0.1 | −799.9 | −779.5 | 40.8 | Non-neutral |
| ASK4 | AlySkp04 | 0.3 | 0.0 | 0.1 | −810.3 | −792.4 | 35.8 | Non-neutral |
| ASK3 | AlySkp04 | 0.3 | 0.0 | 0.1 | −793.3 | −778.7 | 29.1 | Non-neutral |
| ASK1 | AlySkp01 | 0.2 | 0.0 | 0.1 | −759.6 | −745.3 | 28.6 | Non-neutral |
| ASK18 | AlySkp08 | 0.2 | 0.1 | 0.3 | −916.5 | −908.8 | 15.5 | Non-neutral |
| ASK6 | AlySkp02 | 0.3 | 0.1 | 0.3 | −463.8 | −460.0 | 7.5 | Non-neutral |
| ASK11 | AlySkp06 | 0.2 | 0.1 | 0.4 | −776.3 | −773.8 | 5.0 | Non-neutral |
| ASK12 | AlySkp06 | 0.2 | 0.1 | 0.5 | −779.2 | −777.2 | 4.0 | Non-neutral |
| ASK13 | AlySkp13 | 0.2 | 0.1 | 0.6 | −831.1 | −829.8 | 2.6 | Neutral |
| ASK14 | AlySkp15 | 0.3 | 0.2 | 0.6 | −836.6 | −835.5 | 2.2 | Neutral |
| ASK8 | AlySkp09 | 0.2 | 0.3 | 1.6 | −909.0 | −908.0 | 2.0 | Neutral |
| ASK10 | AlySkp09 | 0.2 | 0.2 | 1.4 | −880.8 | −880.4 | 0.9 | Neutral |
| ASK19 | AlySkp03 | 0.2 | 0.2 | 0.8 | −1,110.3 | −1,109.9 | 0.8 | Neutral |
| ASK9 | AlySkp09 | 0.2 | 0.2 | 0.8 | −892.6 | −892.5 | 0.3 | Neutral |
| ASK16 | AlySkp12 | 0.1 | 0.1 | 0.9 | −885.8 | −885.8 | 0.1 | Neutral |
| ASK17 | AlySkp07 | 0.1 | 0.1 | 1.1 | −779.6 | −779.6 | 0.1 | Neutral |
| ASK5 | AlySkp10 | 0.1 | 0.1 | 0.9 | −804.5 | −804.4 | 0.1 | Neutral |
| ASK7 | AlySkp09 | 0.2 | 0.2 | 1.1 | −740.1 | −740.1 | 0.1 | Neutral |

**Notes:**
* $x^2$ ($p = 0.05$, d$f = 1$) = 2.71 (*Yang, 2007*).
Ath, *A. thaliana*; Aly, *A. lyrata*.

constraints than the latter. Interestingly, using GeneWise reannotation (*Birney, Clamp & Durbin, 2004*), we identified a frame-shift mutation in the *ASK6*, *AlySkp02*, and *AhaSkp11* loci that is characteristic of pseudogenes (Figs. S2–S4).

## Expression variation of the *ASK* genes

Low functional constraints do not necessarily mean no function (*Nei & Kumar, 2000*). To understand the functional differences between *ASK* genes, previous studies examined their expression patterns in different tissues/organs by semi-quantitative reverse transcription-PCR (RT-PCR) and in situ hybridization (*Kong et al., 2004*; *Zhao et al., 2003*). The results suggested that all *ASK* genes could be expressed in at least one of six samples examined (mostly in siliques). However, only the highly-expressed *ASK1* and *ASK2* genes showed a consistent result in both studies, while other *ASK* genes with low expression levels were not stably detected by RT-PCR, probably due to technical limitations (*Kong et al., 2004*; *Zhao et al., 2003*). Since then, a number of high throughput transcriptomic analyses, including microarray and RNA-Seq experiments, have been performed in *A. thaliana*, providing good resources to further examine the expression patterns of *ASK* genes in a more statistical manner.

Microarray experiments from 79 *A. thaliana* samples collected from different developmental stages confirmed that *ASK1* and *ASK2* are the two most highly expressed *ASK* genes, followed by *ASK3*. Unlike the previous studies (*Kong et al., 2004*;

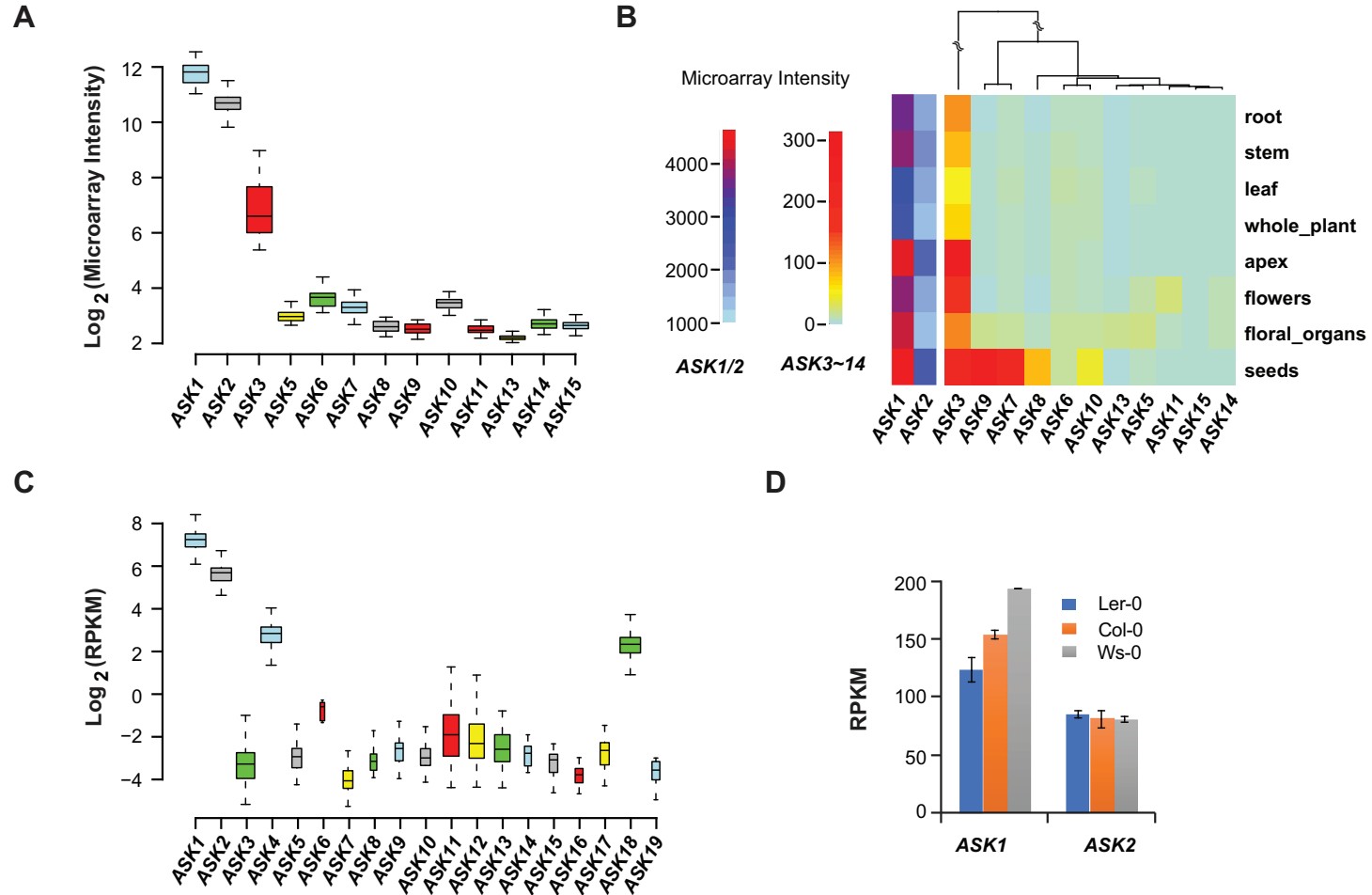

**Figure 2 Expression variation of *ASK* genes.** (A) Variation of absolute expression throughout a developmental time course. (B) A heatmap analysis of mean expression variation of *ASK* genes across eight tissues/organs. (C) Expression variation of *ASK* genes in leaves of 144 accessions. The width of each boxplot is proportional to the number of accessions having a non-zero expression data of the corresponding *ASK* gene. (D) Expression comparison of *ASK1* and *ASK2* in Col-0, Ler-0, and Ws-0.               

*Zhao et al., 2003*), not all *ASK* genes were detected in these microarray transcriptomic analyses, probably due to the low sensitivity and precision of microarray technologies (*Wang, Gerstein & Snyder, 2009*) (Fig. 2A). After comparing the expression levels of *ASK* genes in eight different tissues/organs (79 samples in total), we found that several *ASK* genes, including *ASK1, 2, 3, 7, 8, 9,* and *10*, were expressed most highly in seeds (Fig. 2B), suggesting an important role for SCF-mediated protein ubiquitylation in seed development.

We further analyzed the expression variance of *ASK* genes at the population level. Based on one RNA-Seq experiment (*Schmitz et al., 2013*), which provided a much more precise and sensitive transcriptomic analysis than microarrays (*Wang, Gerstein & Snyder, 2009*), the transcripts of each *ASK* gene could be detected in leaf tissues of 144 *A. thaliana* accessions, but with dramatic variance (Fig. 2C). Consistent with the previous studies and the microarray data, *ASK1* has the highest expression level, followed by *ASK2*. However, the remaining *ASK* genes were only expressed at an average of 1.3 reads per
kilobase of transcript per million mapped reads (RPKM), 121- and 42-fold below the mean expression of *ASK1* and *ASK2*, respectively. Therefore, while *ASK1* and *ASK2* likely have important function(s), the role(s) of the other *ASK* genes in leaves appears minor. Interestingly, the four *ASK* genes (*ASK1*, *2*, *4*, and *18*) whose mean expression levels are significantly higher than the other *ASK* genes (Wilcoxon rank-sum test, $p < 2.2e$-16,) are all under strong evolutionary constraints, and three of them are intronic, further supporting our conclusion regarding the low functional constraints of intronless *ASK* genes.

In addition to the expression variation among different *ASK* genes, the expression of *ASK1* and *ASK2* varied dramatically among different individuals. For example, the highest and the lowest expression levels of *ASK1* were detected in the Gr-1 (480 RPKM, Longitude/Latitude/Altitude = 15.5/47/300) and Co-1 (50 RPKM, Longitude/Latitude/Altitude = −8.3/40.1/100) accessions, respectively, varying by 430 RPKM, while the highest and the lowest expression levels for *ASK2* were detected in Ven-1 (210 RPKM, Longitude/Latitude/Altitude = 5.6/52/parking lot) and Ann-1 (10 RPKM, Longitude/Latitude/Altitude = 6.1/45.9/garden), respectively, differing by 200 RPKM (Fig. 2C). To confirm this result, we also examined the differential expression of *ASK1* and *ASK2* in another RNA-Seq experiment (*Gan et al., 2011*). The expression of *ASK1* varied significantly among Col-0, Ler-0, and Ws-0, which are from distinct geographic regions, while *ASK2* showed mild changes (Fig. 2D). Therefore, although *ASK1* has the highest expression level among all the *ASK* genes, its expression varies the most among individuals, reflecting a possible adaptation of its expression regulation.

## Differential sequence polymorphism of the *ASK* genes

We further examined sequence diversity and polymorphism of the *ASK* genes within the family by comparing number of segregating sites per nucleotide site (θ) and nucleotide diversity (π) values in the regions 500 bp upstream of the start codon, within the CDS, and 500 bp downstream of the stop codon of an *ASK* gene among 774 *A. thaliana* accessions based on their SNP data (*1001 Genomes Consortium, 2016*). We first calculated the allele frequency distribution and noticed that minor allele frequency (MAF) alleles were significantly enriched. In the total populations analyzed, 34% of non-synonymous alleles and 35% synonymous alleles are only present once (i.e., singleton) in the *ASK* family (Fig. 3A), suggesting that many mutations are rare.

Consistent with the enrichment of low MAF alleles, the π values are significantly lower than the θ values in CDSs (Fig. 3B), because the former is determined by allele frequency and the latter is not (*Nei & Kumar, 2000*). Similar to the CDSs, the π values are also significantly lower than the θ values in both upstream and downstream regions, indicating that low MAF alleles are also high in these two regions. As expected, these two regions have higher π and θ values than the CDSs due to their low functional constraints.

The variance of π and θ values within and between different regions of the *ASK* genes indicates different extents of polymorphism. To further examine the evolutionary constraints of polymorphic mutations, a Tajima's *D* value (*Tajima, 1989*) was calculated for the three regions of an *ASK* gene (upstream, CDS, and downstream). According to the
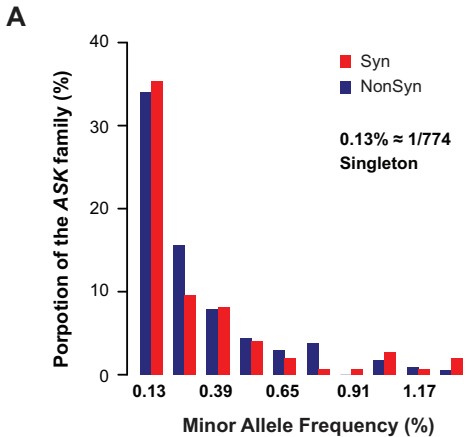

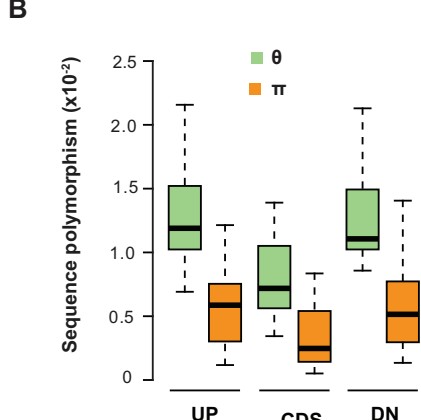

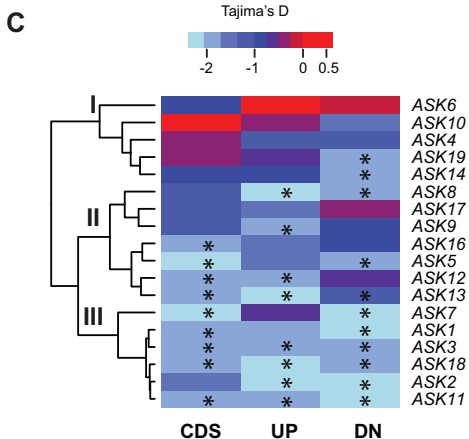

**Figure 3 Sequence polymorphic comparison among different *ASK* genes.** (A) Frequency distribution of rare *ASK* alleles with non-synonymous and synonymous mutations. (B) Variation of sequence polymorphisms in three regions of an *ASK* gene. UP, 500 bp upstream of start codon; CDS, coding sequence; DN, 500 bp downstream of the stop codon. (C) A heatmap representation of Tajima's *D* values demonstrating the differential evolutionary constraints of polymorphic mutations among *ASK* genes. Asterisks indicate a significantly low Tajima's *D* value that deviates from neutral mutations ($p < 0.05$).

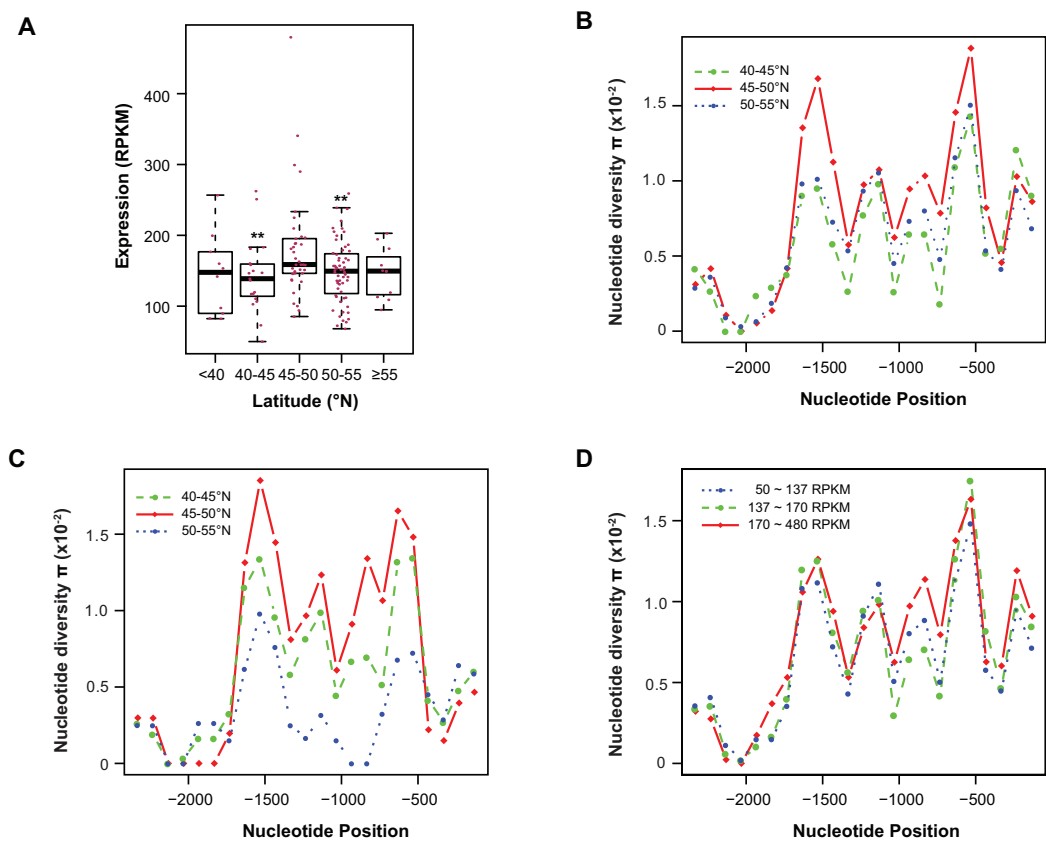

**Figure 4 Contribution of sequence polymorphism to gene expression.** (A) Latitudinal variation of *ASK1* expression. Asterisks indicate that the mean expression of the indicated group is significantly lower than that of the group within latitudes 45–50°N (Wilcoxon rank-sum test, $p < 0.01$). (B) A window-slide polymorphic comparison of a 2.5 kb region upstream of the transcriptional start site of *ASK1* among populations from three latitudinal regions as in (A). (C) A window-slide polymorphic comparison of a group of test populations from three latitudinal regions, performed as in (B). (D) A window-slide polymorphic comparison of three groups of populations with low, medium, and high expression levels of *ASK1* that was performed as in (B).

variance of this value, the 18 *ASK* genes as described in Table 1 were clustered into three groups (Fig. 3C). In Group I, which contains *ASK4*, *6*, *10*, *14*, and *19*, a high Tajima's *D* value was observed in the CDSs, suggesting a balancing or neutral evolutionary process. Group II, which includes *ASK5*, *8*, *9*, *12*, *13*, *16*, and *17*, shows intermediate Tajima's *D* values, some of which are significantly smaller than the Tajima's critical values of neutral mutations ($p < 0.05$) (*Tajima, 1989*), suggestive of purifying selection. The remaining six *ASK* genes (*ASK1*, *2*, *3*, *7*, *11*, and *18*) are clustered into Group III where most have a Tajima's *D* value below the Tajima's critical values of neutral mutations in all three regions ($p < 0.05$). Therefore, mutations in this group are rare, and most sequences are under strong purifying selection. Interestingly, this group enriched five out of eight *ASK* genes that were detected to be under non-neutral changes by orthology comparison (Table 1), further confirming their high evolutionary constrains. It is worth noting that the Tajima's *D* value of the *ASK2* CDS is within the range of Tajima's critical values, suggesting that most SNPs in *ASK2* are neutral.

## Variation association of *ASK1* expression and polymorphism

To associate sequence polymorphism with gene expression variation, we compared the expression patterns and pair-wise nucleotide diversity (π) of 144 *A. thaliana* accessions from the RNA-Seq experiment done by *Schmitz et al. (2013)*. Based on the wide latitudinal distribution of accessions, five subgroups were separated (Fig. 4A). Although no linear regression is observed between individual expression levels and latitudes, accessions within latitudes 45–50°N have significantly higher levels of *ASK1* gene expression than the two flanking regions 5° to the north or south (Wilcoxon rank-sum test, $p < 0.01$). Since 116 out of 144 accessions (81%) reside in these three sub-regions, the populations in these areas were further analyzed. To better understand the relationship between nucleotide diversity and expression levels of *ASK1*, the entire 2.5 kb intergenic sequence upstream of the transcription start site of *ASK1* was compared. A window-sliding analysis (200 bp window and 100 bp slide) showed that the windowed π values from nucleotide −1,600 to −500 are significantly higher in accessions within latitudes 45–50°N than those in the other two regions (Fig. 4B, Wilcoxon rank-sum test, $p < 0.05$). Such a relationship between high polymorphism and high expression suggests that nucleotide variance in the promoter may prevent the binding of a putative transcriptional repressor that may suppress the expression of *ASK1*. To further demonstrate that the accessions within latitudes 45–50°N are highly polymorphic in the promoter region, we also compared accessions from the projects in *Cao et al. (2011)* and *Long et al. (2013)*. In total, 29, 20, and 12 accessions resided in latitudes 40–45, 45–50, and 50–60°N, respectively. Consistently, the accessions from latitudes 45–50°N also have the highest sequence polymorphism from nucleotide −1,600 to −500 among the three groups compared (Fig. 4C). We also applied a different grouping method by separating the aforementioned 144 accessions into low (49 accessions with *ASK1* expression value in the 50–137 RPKM range), medium (50 accessions with *ASK1* expression of 137–170 RPKM), and high (45 accessions with *ASK1* expression of 170–480 RPKM) expression groups. Among these three groups, high nucleotide polymorphism in the −1,000 to −500 region is also most evident in the group with high *ASK1* expression (Fig. 4D).

## Variance of biochemical interactions of ASK1 and ASK2 with known F-box proteins

The different extent of sequence variation between *ASK1* and *ASK2* CDSs (Fig. 3C) led us to speculate that their encoded proteins might show differential strength of interaction with F-box proteins. To address this question, we tested the interactions of ASK1 and ASK2 by pair-wise yeast two-hybrid assay with 15 randomly-selected F-box proteins whose functions have been identified (Fig. 5; Table S2). Due to the identification of neutral polymorphic mutations in the *ASK2* CDS and purifying selection in the *ASK1* CDS, we hypothesized that ASK2 might have lost or reduced its interactions with a number of F-box proteins. To provide a starting point to examine the potential biochemical differences between ASK1 and ASK2, we performed both yeast growth assay on quadruple synthetic dropout medium (SD-Leu-Trp-Ade-His) containing X-α-gal (Figs. 5A–5C) and β-galactosidase activity analysis (Fig. 5D). Interestingly, we detected that two

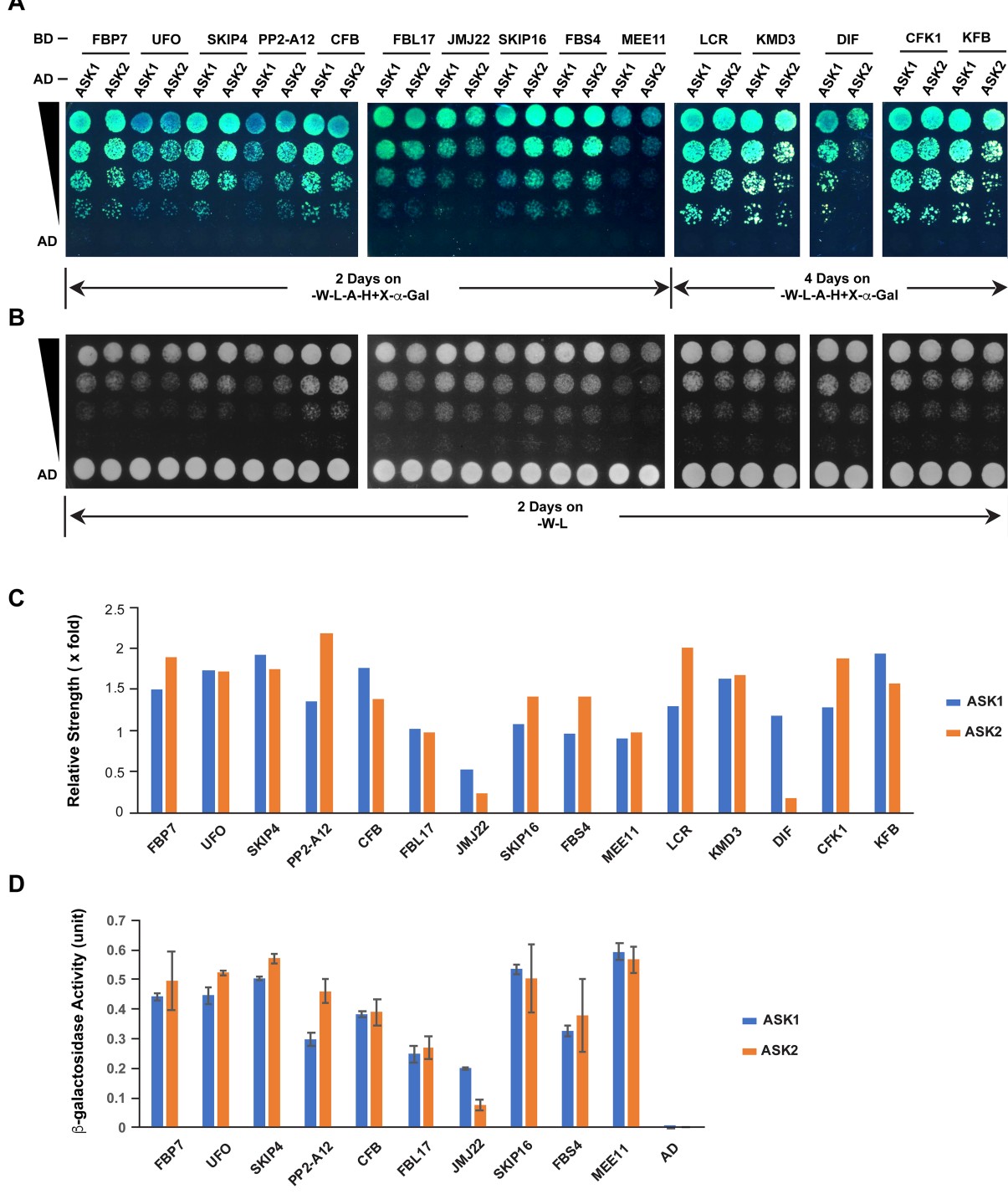

**Figure 5 Quantitative interaction assay of ASK1 and ASK2 with 15 selected known F-box proteins.** The accession IDs of F-box proteins are listed in Table S2. (A) Growth of yeast cells expressing each indicated pair of bait and prey proteins on SD-Leu-Trp-Ade-His+X-alpha-gal medium. (B) Control growth of the corresponding yeast cells in (A) on SD-Leu-Trp medium. Yeast cells were grown on media at 30 °C for an indicated time period and their growth was subsequently recorded by scanning the plates with a Canon 9000F Mark II scanner. (C) Quantification of pairwise ASK1 and ASK2 interactions with 15 F-box proteins as shown in (A). The interaction strength of each pair of bait and prey proteins shown in (A) was normalized by the control growth strength of the corresponding yeast cells in (B). (D) ß-galactosidase activity assay. The mated yeast cells expressing the indicated pair of bait and prey proteins were grown on SD-Leu-Trp medium and used for the assay. The ß-galactosidase activities shown are mean values ± SD measured from two independent assays.

F-box proteins (JMJ22 and DIF) interacted with ASK1 with a strength >2-fold more than with ASK2 (Figs. 5C and 5D). However, the remaining F-box proteins showed an average of 15% variation in their interactions with ASK1 and ASK2, suggesting that the ASK2 protein, albeit carrying recently neutral mutations, still retains the capability in binding many functional F-box proteins. It is worth mentioning that the interaction strengths of ASK1 or ASK2 with different F-box proteins also vary dramatically. This is consistent with the high sequence divergence of *F-box* genes (*Hua et al., 2013*).

## DISCUSSION

Gene duplication has been a long-standing topic of interest in genome evolution (*Ohno, 1970*). In eukaryotic genomes, this process plays an essential role in the expansion of many multi-gene families including those involved in the UPS (*Hua et al., 2011*, *2013*; *Hua, Doroodian & Vu, 2018*; *Hurles, 2004*; *Li et al., 2016*; *Panchy, Lehti-Shiu & Shiu, 2016*). While gene duplication provides the raw genetic material for genome innovation, the large size of multi-gene families has been a hurdle in exploring genome function. For example, it is not yet clear whether and how multiple *ASK* members contributed to evolutionary innovation through sub-functionalization or neo-functionalization, or whether the expansion of the *ASK* family is simply due to the selective advantage of gene dosage or is a result of genomic drift. In this study, we applied several novel approaches to address these questions, as outlined below.

### Retroposed *ASK* genes originated from one single ancestor locus

Instead of cross kingdom long-distant phylogenetic studies (*Kong et al., 2004*, *2007*), we focused on a short evolutionary history within the Arabidopsis genus, so that the orthology relationships and duplication history of individual *ASK* members could be more clearly illustrated. In addition, more advanced sequence alignment tools have been adopted to improve the phylogenetic analysis (Fig. 1). For example, although *Kong et al. (2007)* discovered that *ASK1* and *ASK4* were duplicated through a segmental duplication event by comparing whole genome duplication blocks, their phylogenetic analysis did not reflect this duplication event. Here, we provided improved phylogenetic evidence not only showing the segmental duplication relationship between *ASK1* and *ASK4*, but also demonstrating a single origin of all intronless *ASK* members from the ancestor of *ASK1* through retroposition (Fig. 1). This new discovery is also consistent with the high expression of *ASK1* (Fig. 2) and the short evolutionary history of intronless *ASK* members. Similarly, a previous study on the placental mammalian Gli-Kruppel type zinc finger transcription factor YY1 family revealed that two retroposed intronless subfamilies, YY2 and Reduced Expression 1, were separately clustered but not mingled with the intronic YY1 members (*Kim, Faulk & Kim, 2007*).

### Degenerative processes of *ASK* genes

Since the first draft genome of *A. thaliana* was released, the evolutionary process of both the *F-box* and *ASK* families has been a hot topic in plant biology due to the importance of SCF-mediated protein ubiquitylation and the unequal expansion of the two families.

Although it has been hypothesized that variant ASK proteins might contribute to interactions with a specific group of F-box proteins (*Gagne et al., 2002*; *Kuroda et al., 2012*), in vivo data is currently lacking. Indeed, most functionally characterized F-box proteins physically interact with ASK1 (*Hua & Vierstra, 2011*). Furthermore, recently evolutionary studies have suggested that the number of active *F-box* members is much fewer than the size of the family, due to a genomic drift evolutionary process (*Hua et al., 2011*; *Nozawa, Kawahara & Nei, 2007*; *Xu et al., 2009*). This raised a question as to whether all ASK proteins are involved in the assembly of active SCF complexes.

In this study, we integrated multiple levels of evidence to better describe the functional constraints of individual *ASK* members. Orthology comparisons revealed that 64% of intronless *ASK* genes were under neutral changes indicative of non-functionalization (Table 1). Both developmental and population expression comparisons suggested that most intronless *ASK* members have a very low expression level (Fig. 2). In addition, sequence polymorphism analysis showed a significant enrichment of intronless *ASK* members in the groups whose mutations were under neutral changes (Fig. 3C, clades I and II). Collectively, these data suggest that most, if not all, retroposed *ASK* members are under low functional constraints. This is indeed not surprising, since retroposed genes have a much high rate of pseudogenization, as suggested in the duplication studies of the human genome (*Torrents et al., 2003*). Consistently, our previous evolutionary studies of the *F-box* gene superfamily also discovered a significant enrichment of intronless genes in the pseudogene group (*Hua et al., 2011*). It will be of interest to explore the role of retroposition on the expansion of intronless *F-box* genes.

## The large expression variance of *ASK1* may indicate diverse functions of SCF complexes

Expression comparisons revealed a greater than 430 RPKM (~10-fold) variance in *ASK1* transcript levels among individual *A. thaliana* accessions (Fig. 2). Sequence analyses further suggested the presence of a putative transcriptional repressor that might contribute to such large variations in expression. Upstream sequences with more variable sites may prevent the binding of this transcriptional repressor, thus increasing expression. The finding that more polymorphic upstream sequences result in higher expression may support this model (Fig. 4). Such large variation in *ASK1* expression among natural variants suggests that ASK1 is either very effective in promoting the assembly of SCF complexes or is involved in an as-yet-unknown-mechanism to regulate the polymorphic functions of SCF complexes. It will be noteworthy to further investigate the proteomic variance of SCF-mediated protein ubiquitylation in the future.

## CONCLUSIONS

In this study, our improved phylogenetic analysis resolved the inconsistency between the phylogeny of *ASK* genes and the single origin of retroposed *ASK* members (*Kong et al., 2007*). Through evolutionary selection analysis and sequence polymorphism comparison, we discovered both adaptive and degenerative evolutionary processes in the *ASK* family. Yeast two-hybrid quantitative interaction assay and expression analysis

across different accessions further indicated that recent neutral changes in the *ASK2* CDS likely weakened its interactions with F-box proteins and that highly polymorphic upstream regions of *ASK1* may contribute to adaptive roles of SCF complexes in Arabidopsis, respectively.

### Funding

This work was supported by a National Science Foundation CAREER award (MCB-1750361 to Zhihua Hua) and a Baker Award from Ohio University (IA1017002 to Zhihua Hua). Zhenyu Gao was a senior visiting scholar in the Hua lab, in part supported by 151 Talents Project from the province of Zhejiang, China. The funders had no role in study design, data collection and analysis, decision to publish, or preparation of the manuscript.

### Grant Disclosures

The following grant information was disclosed by the authors:
National Science Foundation CAREER award: MCB-1750361.
Baker Award from Ohio University: IA1017002.
Hua lab, in part supported by 151 Talents Project from the province of Zhejiang, China.

### Competing Interests

The authors declare that they have no competing interests.

### Author Contributions

- Zhihua Hua conceived and designed the experiments, performed the experiments, analyzed the data, contributed reagents/materials/analysis tools, prepared figures and/or tables, authored or reviewed drafts of the paper, approved the final draft.
- Zhenyu Gao performed the experiments, revised the manuscript.

### Data Availability

The raw data are available in File S1 and Table S3.

### Supplemental Information

Supplemental information for this article can be found online at http://dx.doi.org/10.7717/peerj.6740#supplemental-information.

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
