# Peer review of "Adaptive and degenerative evolution of the S-Phase Kinase-Associated Protein 1-Like family in Arabidopsis thaliana"

_PeerJ, doi:10.7717/peerj.6740_

## Round 0.1 · original submission · Minor Revisions

Please carefully address all the critical issues pointed by both reviewers and revise your manuscript accordingly.

Reviewer 1 ·

Basic reporting

Overall the structure of the paper is well-organized and supporting literature is well referenced. The figures are high-quality, clear, relevant and are well labelled.

Experimental design

The experimental design is clear and the authors clearly stated their hypothesis and the results presented by the authors are rigorous.

Validity of the findings

No comment.

Additional comments

I only have a few minor comments for the authors:
1. For Figure 2A, since the the microarray intensities for ASK1 and ASK2 are roughly three logs higher than ASK3~14, it is better to break the Y-axis at 0.1 to visualize the expression level of ASK3~14 better. By using a continuous y-axis, the expression value of the ASK3~14 appears to be zero, which is not the case. The same is true for Figure 2C. I recommend to break the Y-axis at 50 to better visualize the RPKM for ASK3~14.

2. For Figure 2C and 2D, please change the Y-axis label to "RPKM". RPKM stands for Reads Per Kilobase of transcript, per Million mapped reads. RPKM does not mean the counts for all RNA.

3. For Figure 4A, latitudinal variation of ASK1 expression, the left panel is redundant. I would suggest to remove the left panel and keep the right panel for 4A.

·

Basic reporting

The overall structure of the manuscript is appropriate. The Introduction gives a good basic description of the ubiquitin-26S proteasome system and gives, for the most part, sufficient background information about SCF complexes (see comments below). The rationale for the work is clearly presented. The Results and Discussion sections are well-organized, logical, and, for the most part, easy to follow (with the exception of some grammar and phrasing issues; see below). The Materials and Methods section gives sufficient details of the analyses. The figures are, for the most part, clear and well-composed.

Some of the grammar and phrasing is a bit rough and there are several typographical errors I noticed. The manuscript should be carefully checked for these issues. I give some examples here, but I have not highlighted every problem I saw.

Specific critiques:

1. The second sentence of the Abstract (lines 31-33) is structured a bit confusingly. Do the authors mean that genome size is extremely varied, and that there is a great deal of polymorphism seen in large gene families?

2. Line 36: "first provided" could be "provide". Also, I am not sure what "polymorphic evidence" is.

3. Line 37: "of" could be "in".

4. Lines 48, 49: I don't believe that the second sentence of the Introduction (about proteins being involved in mechanical movement) adds anything important.

5. Line 102: "the rapid evolutionary process" could be " rapid evolution".

6. Since the fourth paragraph of the Introduction mentions the Type II Skp1 genes, it would be good to say something about the differences between Type I and II Skp1 genes in the Introduction.

7. Lines 111-113: The "These datasets provide a unique opportunity to……" sentence is rough. Maybe "…fine-tune the phylogenetic relationships and fixation processes of rapidly-evolving genes in plants, which may help better define their functional constraints.".

8. Lines 113-116. I don't understand the "Because the size of the Skp1…." sentence. What is the connection between the smaller size of the Skp1 gene family and focusing the analysis on Arabidopsis?

9. Line 180: "…towards each species…" could be "…between species…".

10. Line 182: The text says 18 common ancestors of the Arabidopsis Skp1 genes, but Figure 1C shows 19.

11. The caption for Figure 1: "Statistical significance equal or greater than 50%....". Also (line 10), "a Skp1 gene..", not "an".

12. The graph in figure 2C clearly shows the variation in expression between different thaliana accessions for ASK1 and ASK2, but the y-axis scale makes it difficult to see if other ASK genes also show much variation. The Results text (lines 229-231) suggests that those ASK genes do, but it's hard to see as the data is presented.

13. Figure 3 caption (line 553): "… deviates from neutral mutations…"

14. Is Table S1 missing ASK6 for the OrthoMCL12 group? Also, the Aha and Aly sequences seem to have been switched for OrthoMCL groups 4-13.

15. The ASK18, AlySkp08, and AhaSkp07 protein sequences in the alignment in Figure S1 all are shown to contain 22 amino acids N-terminal to a methionine that corresponds to the start methionine in the other Skp sequences. I checked Phytozome and TAIR and, in both, the ASK18 protein is predicted to start with the methionine. Is this a mistake? Did the authors change the annotation?

Experimental design

Previous publications have used bioinformatic approaches to explore the Skp1 family in the plant kingdom (ex. Kong et al., 2007). What is new in this manuscript is analysis of the family in three closely related Arabidopsis species and in multiple accessions of Arabidopsis thaliana, allowing for investigation of evolution of the family over a shorter evolutionary time-span. A tree based on more closely-related sequences allows the authors to better pinpoint evolutionary events that gave rise to the family variation seen in Arabidopsis. As an example, the tree generated by the authors allows them to speculate that a single retroposition event gave rise to all current intron-less Skp1 genes in Arabidopsis. Given this data, the authors are able to analyze the evolutionary constraints on the ASK genes and the degree of sequence polymorphism in them and correlate those findings with gene structure (intron-containing vs. intron-less). The authors also take advantage of recent gene expression studies to explore the expression of the ASK genes in more detail than was previously possible. Collectively, the results presented here do contribute new knowledge to the field.

The question whether many of the ASKs are functional or, if they are, how important they are to SCF function, is one that has been present in the field since the gene family was first characterized. The work done here cannot directly answer those questions, but I believe the analyses and data presented do contribute useful new data that will help the field as these questions are further pursued.

I believe the Materials and Methods section does sufficiently explain the methodologies used.

Validity of the findings

Overall, I agree with the authors in their general conclusions from the data, though I do have some questions as well as concerns about some specific interpretations/conclusions. See the comments following:

1. The authors appear to claim that most Arabidopsis Skp1 genes duplicated in a 5-10 mya time-frame, based on the fact there are distinct clades with sequences from all three species and the timing of when the species split (Lines 164-166). I am confused about this. Couldn't the duplications have occurred significantly earlier than 5-10 mya and produced the same type of groupings in the tree? Is there some additional piece of information I am missing?

2. Lines 169-170: Do the authors mean that the three ancestor loci were produced by retroposition and then each was further duplicated by tandem duplication? The tandem duplication events indicated on the tree (Figure 1B) occur after the a, c, and d loci arose.

3. Several predicted Skp1 proteins (e.g. ASK6, AK7, AlySkp14, AhaSkp6) lack sequence seen in other family members. In my experience this can happen when there are frameshifts or in-frame stop codons in (otherwise normal-looking) coding regions. Gene structure prediction algorithms sometimes attempt to avoid these (by predicting an intron that skips the coding sequence over the frameshift or stop, for instance), leading to coding regions that are predicted to encode a protein lacking sequence that is present in all other family members. Could the same be happening here and are the genes apparently encoding these proteins actually pseudogenes? Is there genomic sequence present that appears to encode the missing parts of the proteins?

4. Partial sequences for Skp1 genes can be found in the Arabidopsis thaliana genome (Kong, 2007). Did the authors encounter similar partial sequences in the lyrata and halleri genomes? I'm particularly interested in the halleri genome, as the authors calculated substantial loss of Skp1 genes since the split from lyrata. Is there any residue of those lost genes in the halleri genome? I'd be surprised that there is so much sequence change that those genes are no longer recognizable. Is the likely mechanism of loss complete deletion?

5. I would temper the statement that high expression of selected ASK genes in seeds (lines 224-225) indicates an important role for SCF complexes in seed development. That may be the case, but I'm not sure that high expression alone confirms it.

6. The connection between increased nucleotide polymorphism in the -500 to -1600 region and increased expression in ASK1 seems tenuous to me. While the expression of ASK1 in the group of accessions from the 45-50 latitudes is statistically significantly higher than the 40-45 and 50-55 groups, the absolute difference seems to be small. There is also a lot of variation among the individual accessions. Is there some more direct way to compare nucleotide diversity with expression, without using the latitude groups as an intermediary? Can you bin the accessions based on expression level (say, high, medium, low) and then compare nucleotide diversity between those groups? The latitude binning strikes me as arbitrary.

7 I'm not sure that testing a single ASK1 and a single ASK2 for F-box interactions really allows for the conclusion that neutral polymorphic mutations in ASK2 genes do not significantly affect the F-box/ASK interaction. It seems to me that you need to test a group of them to really show that. Any one ASK2 may by chance have mutations that do not affect the strength of the interaction even if, as a group, the relaxation of selection pressure results in a greater likelihood of loss of F-box interaction. It seems the best experiment would be to test a series of both ASK1 and ASK2 proteins from the different accessions and show that the greater variability in the ASK2 proteins does not affect these interactions. Perhaps I am not understanding the author's reasoning correctly. I do realize the large amount of work testing more ASKs would entail. Perhaps the authors could moderate their conclusion by stating that their Y2H analyses are a starting point for testing this hypothesis.

8. I'm not sure I agree with the heading in the Discussion (lines 395-396) that the large variation in ASK1 expression between accessions is indicative of varying function. It could simply mean that there is a great deal of flexibility for ASK1 expression to change and still have it perform its functions. The authors do seem address this possibility in the body of the paragraph (lines 403-404: "…that ASK1 is either very effective assisting the assembly of the SCF complex…").

---

## Round 0.2 · accepted · Accept

Since all the critiques of both reviewers were addressed and the manuscript was revised accordingly, I am pleased to accept this manuscript for publication in its present form.

#